# Gut Microbiota and B Cell Receptor (BCR) Inhibitors for the Treatment of Chronic Lymphocytic Leukemia: Is Biodiversity Correlated with Clinical Response or Immune-Related Adverse Event Occurrence? A Cross-Sectional Study

**DOI:** 10.3390/microorganisms11051305

**Published:** 2023-05-17

**Authors:** Valentina Zuccaro, Greta Petazzoni, Irene Mileto, Marta Corbella, Erika Asperges, Paolo Sacchi, Sara Rattotti, Marzia Varettoni, Irene Defrancesco, Patrizia Cambieri, Fausto Baldanti, Luca Arcaini, Raffaele Bruno

**Affiliations:** 1Division of Infectious Diseases I, Fondazione IRCCS Policlinico San Matteo, 27100 Pavia, Italy; 2Microbiology and Virology Department, Fondazione IRCCS Policlinico San Matteo, 27100 Pavia, Italy; 3Department of Medical, Surgical, Diagnostic and Pediatric Science, University of Pavia, 27100 Pavia, Italy; 4Division of Hematology, Fondazione IRCCS Policlinico San Matteo, 27100 Pavia, Italy; 5Department of Molecular Medicine, University of Pavia, 27100 Pavia, Italy

**Keywords:** gut microbiota, B cell receptor inhibitors, chronic lymphocytic leukemia, clinical response

## Abstract

Several studies have strengthened the link between the gut microbiota (GM) and the response to immunotherapy in patients with tumors, highlighting the potential role of GM as a biomarker of response. Targeted therapies including B-cell receptor (BCR) inhibitors (BCRi) represent the newest approach to the treatment of chronic lymphocytic leukemia (CLL); however, not all patients achieve a satisfactory response, and immune-related adverse events (irAEs) can also impact the efficacy. The aim of the study was to compare GM biodiversity in patients with CLL, treated with BCRi for at least 12 months. Twelve patients were enrolled: 10 patients in the *responder* group (R) and 2 patients in the *non-responder* group (NR). We identified seven patients (58.3%) who experienced adverse reactions (AE). Although we did not observe a significant difference across the study population in terms of relative abundance and alpha and beta diversity, we found a differing distribution of bacterial taxa between the analyzed groups. We noted a higher level of the class Bacteroidia and the order Bacteroidales in the R group, and an inversion in the Firmicutes and Bacteroidetes ratio in the AE group. No prior studies have focused on linking GM and response to BCRi in these patients. Although the analyses are preliminary, they provide suggestions to guide future research.

## 1. Introduction

Chronic lymphoproliferative disorders (LPD) are a large group of diseases characterized by the monoclonal expansion of lymphocytes leading to monoclonal lymphocytosis, lymphadenopathy and bone marrow infiltration [1]. Among LPD, chronic lymphocytic leukemia (CLL) is the most common leukemia observed in adults. According to the 2018 International Workshop on Chronic Lymphocytic Leukemia (iwCLL) updated consensus [2], at the time of CLL diagnosis, treatment is indicated only for active disease (i.e., peripheral cytopenias due to bone marrow failure, massive or symptomatic lymphadenopathy or splenomegaly, progressive lymphocytosis or disease-related symptoms). Many different molecules can be used, and more recently, targeted therapy for CLL has dramatically reshaped the landscape of treatment. For example, Bruton tyrosine kinase (BTK) inhibitor-based therapies have demonstrated that these compounds are highly effective in disease control [3]. Ibrutinib, the first-in-class inhibitor of Bruton tyrosine kinase (Btk-inhibitor), and Idelalisib, the first-in-class inhibitor of phosphatidylinositol 3-kinase δ (PI3K-inhibitor), have been approved for the treatment of indolent B-cell malignancies. At the time of our study enrollment, these drugs were used especially used for previously unmet needs (i.e., patients with relapsed or refractory disease, high-risk cytogenetic or molecular abnormalities, or with comorbidities) [4]. Despite the high efficacy of BCRi, a portion of patients still do not respond to therapy or experience immune-related adverse events (irAEs) [2]. Since most irAEs of these new molecules are of gastrointestinal origin (e.g., colitis and diarrhea), researchers have suggested a potential role of the gut microbiota (GM) in modulating the response to these agents and in the development of such adverse reactions. Robust evidence regarding the cross talk of GM and host immune responses in melanoma patients has also accumulated, suggesting the potential involvement of the GM in modulating the efficacy of such anti-CTLA4- and anti-PD1-based therapies. The findings of a preclinical study by Vetizou et al. showed an association between the relative abundance of *Bacteroides fragilis* and *Burkholderia cepacia* and the efficacy of anti-CTLA4 antibodies in reducing sarcoma tumor growth in mice [5]. Following this, metagenomic analysis of melanoma patients’ fecal gut microbiomes showed a higher relative abundance of *Clostridiales*, in particular those belonging to the *Ruminococcaceae* and *Faecalibacterium* families in anti-PD1 responders when compared to non-responders to therapy [6]. Even in hematological disorders, several studies have strengthened the bidirectional link between GM and the response to therapy: GM composition can influence drug efficacy and, conversely, chemotherapeutics can alter GM [7].

Although a lack of consensus exists, convincing evidence has increased, such as the findings of Hakim et al. regarding a correlation between GM composition and the development of chemotherapy-related adverse effects in patients affected by acute lymphocytic leukemia. For instance, the relative abundance of *Enterococcaceae* or *Streptococcaceae* is associated with infections, and that of *Proteobacteria* is associated with febrile neutropenia [8]. With this in mind, it is possible to consider GM composition as a biomarker of response and of adverse events [9] and, at the same time, to imagine the manipulation of gut microbiota composition as a new strategy to enhance the response to treatment. Concerning this, Baruch et al. noted in a phase I study (NCT03353402) a longer progression-free survival (PFS) in refractory metastatic melanoma patients who underwent fecal microbiota transplantation from a complete responder patient [10,11]. So far, no study has investigated the alteration of GM in patients diagnosed with CLL treated with BCRi and the potential association with therapy outcome and irAEs. We hypothesized that the immune-stimulatory and antitumor effects of BCRi (Btk- and PI3K-inhibitors, such as Ibrutinib and Idelalisib) in patients with CLL could be influenced by distinct gut microbiota compositions. Hence, the aim of this study was to investigate whether GM characteristics were correlated with clinical responses and the occurrence of irAEs in patients affected by CLL who underwent treatment with BCRi.

## 2. Materials and Methods

### 2.1. Study Setting

We designed our research as a cross-sectional study. All patients from the Hematology Unit of Fondazione “IRCCS Policlinico San Matteo di Pavia” affected by CLL who started BCRi therapy with Btk- or PI3K-inhibitors at least 12 months before the date of study enrollment were enrolled. We collected fecal samples in those patients at their 12-month visit after initiation of BCRi treatment. For patients whose BCRi was discontinued for any reason before the 12-month visit, a visit nearest the 12th month (±1 month) was scheduled in order to collect data. Fecal samples were examined with next-generation sequencing (NGS) techniques. All participating patients signed an informed consent form. We split the study population into a responder group (R) and a not-responder group (NR) according to the response definition of the 2018 iwCLL updated consensus as follows:

R Group: Patients meeting the clinical response criteria (no evidence of clinical progression criteria) and undergoing ongoing therapy with BCRi.NR Group: Patients with clinical progression criteria and BCRi therapy discontinuation within 12 months of treatment initiation.

We further collected demographic data such as age, sex, clinical history, intake of antibiotic therapies during the last 3 months, hematological history (date of diagnosis of CLL, cytogenetic and molecular features, treatment lines, clinical and non-clinical response criteria, adverse events), patients’ outcome, and cause of death. Special diets, such as celiac, lactose or casein-free, were also recorded. Response assessment was evaluated at least 2 months after the beginning of treatment and thereafter, as is clinical practice.

We further categorized the study population into two other groups: the AE group, patients who experienced irAEs at any time during the treatment, and the NAE group, patients who did not experience irAEs. The study was conducted according to the Strengthening the Reporting of Observational Studies in Epidemiology (STROBE) statement for reporting observational studies [12], and was approved by the local Ethics Committee (Comitato Etico Area Pavia) and Institutional Review Board (P-20180107349, 7 December 2018).

### 2.2. Objectives

The primary objective of this study was to examine the GM composition in a population of patients with CLL in order to recognize differences between responders (R) and non-responders (NR). The secondary objective was to assess GM biodiversity in patients with/without irAEs.

### 2.3. Biological Samples and DNA Sequencing

At the first visit falling into the study period (at least 12 months from BRCi initation), patients were instructed to provide a stool sample at the following visit. All stool samples were collected and stored at −80 °C in the Laboratory of Microbiology and Virology Unit of Fondazione IRCCS Policlinico San Matteo (Pavia, Italy). Then, we extracted the genomic DNA using the DNeasy^®^ PowerSoil Pro Kit (Qiagen, Hilden, DE, USA) according to the manufacturer’s instructions and the V3-V6 hypervariable regions of the 16S rRNA gene were amplified using Microbiota Solution B Kit (Arrow Diagnostics s.r.l., Genova, Italy). Extracted DNA was finally sequenced using a paired-end 2 × 250 bp cycle on the Illumina MiSeq sequencing system (Illumina, CA, USA).

### 2.4. Statistical and Bioinformatic Analyses

Raw reads were processed with the MicrobAT (Microbiota Analysis Tool) system of the SmartSeq S.r.l. (Novara, Italy). In particular, the software filtered row reads for length and quality (data quality evaluation), then it aligned them against the RDP database (vr. 11.4) and it assigned them to a species if they met specific criteria (query coverage ≥80% and similarity ≥97%). Thus, the software produced the phylum, class, order, family, genus, and species level for each bacterium found in the samples.

Results were then processed using a dedicated pipeline in R software. Firstly, data were processed in order to identify and remove features that were unlikely to be useful when modeling the data. Features were filtered for low count, considering features with at least 4 counts in 20% of the samples (20% of prevalence cut-off), and low variance (based on the interquartile range). Finally, filtered features were used to compute the relative abundances of microbial taxa in each sample. Microbial profiles of taxa with a prevalence of at least >1% (most abundant) in one sample of the dataset were compared between groups: R vs. NR group and AE vs. NAE group, using the Mann–Whitney U-test. The significance threshold (*p*-value) was set at 0.05.

The within-sample diversity (α-diversity) was estimated at each taxonomic level using the phyloseq package in R [13]. The observed richness and Shannon indexes were compared between groups using the Mann–Whitney U-test (*p*-value threshold set at 0.05).

The diversity in composition among samples (β-diversity) was evaluated at all taxonomic levels, computing the dissimilarity matrices with the vegan package in R [14]. Bray–Curtis distances were applied and then visualized as principal coordinate analysis (PCoA). To statistically assess the differences between groups, permutational multivariate analysis of variance (PERMANOVA) was performed with the threshold for *p*-value set at 0.05.

## 3. Results

Thirteen patients were enrolled from January 2019 to October 2019 in the Hematology Unit of Fondazione “IRCCS Policlinico San Matteo di Pavia”. One was further excluded. The average age of patients was 63.6 years old, and 50% (6/12) were male. No patients reported a special diet. The hematologic diagnosis was CLL in all patients. Seven patients (58.3%) were treated with Ibrutinib and five (41.7%) were treated with Idelalisib. Only one patient was treated with BCRi as the first line of treatment. For all other patients, BCRi represented a later course of therapy.

According to the response definition of the 2018 iwCLL updated consensus [2], we split the study population into the *responder* group (R) and *non-responder* group (NR). We identified 10 patients who met the clinical response criteria (R) and 2 patients who met the clinical progression criteria (NR). Moreover, we identified seven patients (58.3%) who experienced adverse reactions (AE group), mainly diarrhea (three out of seven patients), followed by fever and liver toxicity. Of those patients, five (71.7%) were taking Idelalisib. Interestingly, the two NR patients were taking Ibrutinib. Although the two NR patients experienced irAEs, these were not the leading cause of discontinuation of the treatment. The demographic and hematological features of the study population are reported in Table 1.

### 3.1. Taxonomic Composition and Diversity of Fecal Bacterial Communities in the Responder and Non-Responder Group

We examined the gut microbial signature across the *responder* and *non-responder* groups. The average relative abundance of bacterial phyla, classes and orders between the two groups did not statistically differ, as shown in Figure 1. Concerning the phylum, the frequency of *Firmicutes* was 38.3% in R and 47.3% in NR, while that of *Bacteroidetes* was 35.7% and 28.2%, respectively. Although there was no significant difference, at the class level we found lower levels of fecal *Clostridia* (32.3%) and higher levels of fecal *Bacteroidia* (34.9%) in the R group, while the frequencies in the NR group were the opposite (40.9% and 27.9%, respectively) (Figure 1b). Regarding the order, despite the lack of statistical significance, we found a different average relative abundance between the two groups: *Bacteroidales* were the most abundant in the R group (34.9%) as *Clostridiales* were the most abundant in the NR group (40.1%) (Figure 1c).

As for the species, the abundance of *Bacteroides* sp. S-18 was higher in the NR group compared with the R group (*p* = 0.034, Mann–Whitney U-test).

Only Ibrutinib-treated patients were selected for GM analysis to provide a homogeneous sample. Notably, the NR group only received Ibrutinib, while the responder group within the Ibrutinib-treated cohort (RI) differed. The taxonomic composition of the RI cohort did not differ compared to the first one. At the phylum level, the *Firmicutes* frequency was 32.2% in RI, while that of *Bacteroidetes* was 46.3%. Similarly to the R group, at the class level, lower levels of fecal *Clostridia* (32.3%) and higher levels of fecal *Bacteroidia* (34.9%) were confirmed in the RI group when compared to the NR group (Appendix A). Concerning the order, the trend was confirmed, and we found a greater average relative abundance for the RI group compared with the NR group: *Bacteroidales* were the most abundant in the RI group (44.9%). No significant difference was found between RI and NR patients in terms of α-diversity and β-diversity (Appendix A).

#### α-Diversity and β-Diversity Statistical Analyses in the Responder and Non-Responder Groups

The α-diversity indexes (observed richness and Shannon) were computed at all taxonomic levels to analyze the within-sample diversity, and there were no significant differences among the groups (Figure 2).

The β-diversity (diversity in composition among samples) was computed at all taxonomic levels, and we did not observe any significant differences, suggesting the lack of a different gut microbial community between the analyzed groups (Figure 3).

### 3.2. Taxonomic Composition and Diversity of Fecal Bacterial Communities between Patients with and without irAEs

The study population was later categorized according to the occurrence of irAEs: the AE group for patients who experienced irAEs and the non-adverse events (NAE) group for patients who did not experience such adverse events. Despite the lack of statistical significance, at the taxonomic rank of phylum, we observed a different average relative abundance across the AE group and NAE group. Specifically, we found a higher number of fecal *Firmicutes* (45.7%) and a lower number of fecal *Bacteroidetes* (26.1%) in the AE group, while the frequencies in the NAE group were the opposite (31.5% and 46.1%, respectively) (Figure 4a). Similarly, at the class level, the abundance of fecal communities differed between the two groups without statistical significance: *Clostridia* was the most abundant in the AE group (41%), while *Bacteroidia* was the most represented in NAE group (44.7%) (Figure 4b). At the order level, we found different average relative abundances between the two groups: *Bacteroidales* were the most abundant in the NAE group (44.7%), while *Clostridiales* were higher in the AE group (40.9%) (Figure 4c).

Interestingly, some bacterial taxa were only found in one of the two groups, as shown in Table 2.

#### α-Diversity and β-Diversity Statistical Analyses in Patients with or without irAEs

α-diversity indexes were also calculated according to irAE categorization, and we did not observe any significant differences between the two groups for all of the taxonomic ranks (Figure 5).

Similarly, we performed the between-sample diversity (β-diversity) evaluation, and the data suggested no significant differences in all bacterial consortia when comparing the samples according to irAE categorization (Figure 6).

## 4. Discussion

In this cross-sectional exploratory investigation, we reported an overall analysis of the gut microbiota of patients with CLL receiving BCRi treatment, and we speculated the that GM signature was associated with clinical response and the occurrence of irAEs. Although we did not observe a significant difference across the study population in terms of relative abundance and alpha and beta diversity, we found a differing distribution of bacterial taxa between the analyzed groups. In particular, we noted a higher number of bacteria from the class *Bacteroidia* and the order *Bacteroidales* in the R group when compared to NR, who are conversely enriched in the class Clostridia and the order *Clostridiales.* Similarly, in the AE group, we observed a higher number of bacteria from the class *Clostridia* and the order *Clostridiales*. These findings were consistent with data reported by Karpinets et al. about a population affected by cervical cancers, showing that a fecal community rich in *Clostridia* was characterized by pathways involved in stress response [15]. Not surprisingly, the emerging view based on numerous studies is that collective interactions within the bacterial community drive disease through the dysregulation of mucosal immunity and the disruption of the mucosal barrier. In the AE group, we also found an inversion in the *Firmicutes* and *Bacteroidetes* ratio. As is widely known, GM produces bio-products, and short chain fatty acids (SCFAs), such as acetate, propionate and butyrate, are the most relevant of these, such as in inflammation pathways. Butyrate, mainly produced by the phylum *Firmicutes*, provides for the maintenance of the integrity of colonocytes and plays a key role in the control of inflammation [16]. With this in mind, the lower abundance of *Firmicutes*, which means lower butyrate production, is not an unexpected finding in the AE group. Among *Firmicutes*, the butyrate-producing *Faecalibacterium prausnitzii* had been widely proved to be associated with a good response to immune checkpoint inhibitors (ICIs) and a higher risk of developing irAEs (in particular, immune-related colitis) in metastatic melanoma patients [17]. On the other hand, among *Bacteroidetes*, *Akkermansia muciniphila* seemingly plays a protective role in the occurrence of colitis [16]. In agreement with literature data, in our cohort, we did not find *Akkermansia muciniphila* in the AE group and we failed to demonstrate the association between *Faecalibacterium prausnitzii* and irAEs; conversely, we found this species only in the NAE group. An explanation of these contradictory data could be the nonhomogeneity of irAEs other than colitis that we observed. Moreover, we speculated that the metabolites secreted by *F. prausnitzii*, blocking NF-κB activation and IL-8 secretion [18], might play a role in attenuating the manifestation of irAEs, as demonstrated by Liu et al. in patients treated with anti-PD-1 inhibitors: a higher abundance of *Faecalibacterium* was associated with mild irAEs [19]. Several confounding factors might further impact the interpretability of our findings, especially the study design and the small sample size. As discussed above, we found different signatures across the study population, and we tried to understand which factors could drive these differences and how. Firstly, most of the study population had already received chemotherapy and many of them additionally received antibiotic prophylaxis. In this regard, the main challenge is to assess the impact of antibiotics in modulating GM, and then the antitumor effect. We already mentioned that chemotherapeutics can alter GM; for example, in preclinical studies, cyclophosphamide altered GM composition through the selective translocation of species of Gram-positive bacteria into secondary lymphoid organs, leading to tumor invasion of pathogenic TH17 cells [7,20]. With this in mind, in a large German cohort of patients affected by CLL and treated with cyclophosphamide, according to Viad’s findings, Pflug et al. showed that treatment with anti-Gram-positive reduced anticancer efficacy [21].

As discussed above, our study has several limitations that might impact the interpretation of our findings. Furthermore, the small sample size prevented us from reaching statistical significance. Further validation of our findings in prospective cohorts or analyses conducted in association with randomized controlled trials should be promoted. Despite these limitations, the strength of our study comes from a strict definition of patient characteristics, and, to the best of our knowledge, no previous research on this topic is available.

## 5. Conclusions

In summary, despite the lack of statistical significance, we found a different distribution of bacterial taxa between the analyzed groups, and our findings are consistent with the currently well-known evidence regarding the cross-talk of GM and host immune responses in melanoma patients. In the era of precision medicine, further studies in this group of patients are needed in order to better define the prognostic role of the GM signature and to develop targeted interventions such as the manipulation of GM through fecal microbial transplantation.

## Figures and Tables

**Figure 1 microorganisms-11-01305-f001:**
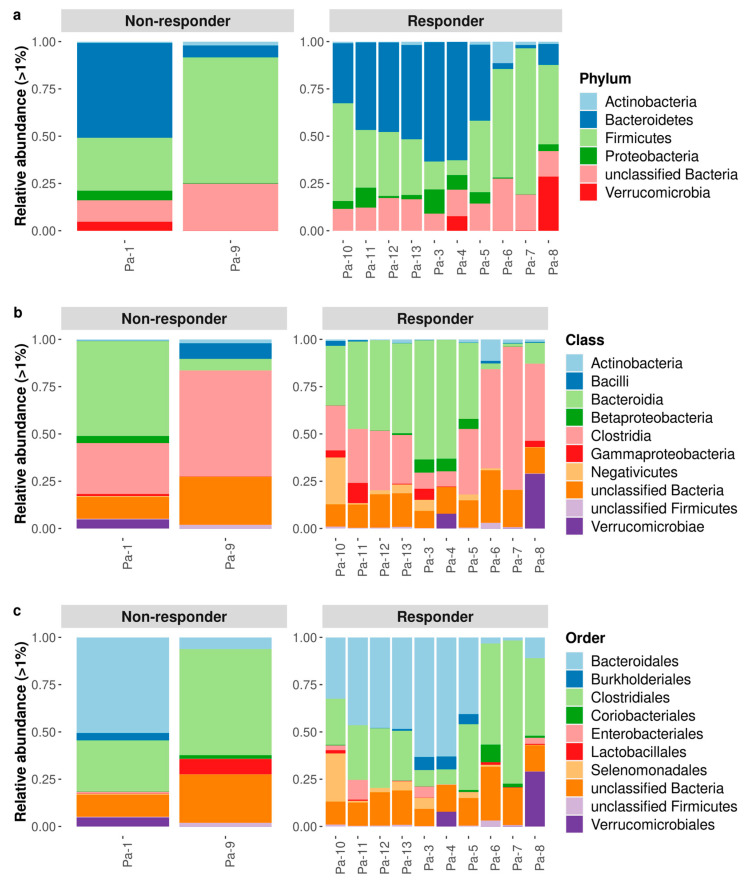
Taxonomic composition and diversity of fecal bacterial communities for each patient stratified for responder and non-responder group. The most represented phyla (**a**), classes (**b**), and orders (**c**) identified in the study groups are shown with relative abundance. Only taxa whose relative abundance was >1% in at least one group were included.

**Figure 2 microorganisms-11-01305-f002:**
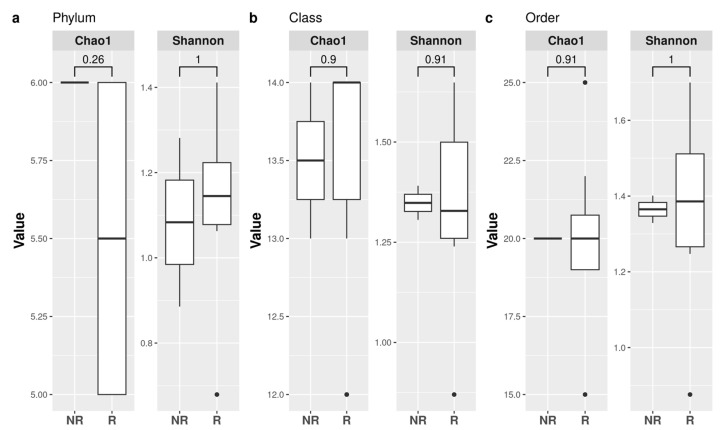
Alpha-diversity. Observed richness and Shannon indices are presented at the taxonomic level of phylum (**a**), class (**b**), and order (**c**). Significant (*p* < 0.05) comparisons between responder (R) and non-responder (NR) patients are indicated in the boxplot.

**Figure 3 microorganisms-11-01305-f003:**
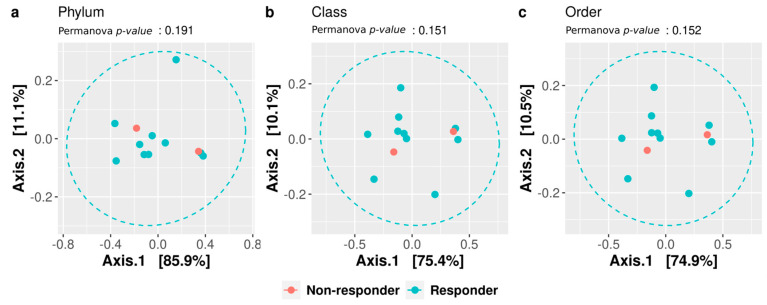
Beta-diversity. The microbiota distances were evaluated through the Bray-Curtis dissimilarity matrix at the taxonomic level of phylum (**a**), class (**b**), and order (**c**) and visualized through principal coordinate analysis (PCoA). Each point represents the microbiota composition of one sample stratified for outcome (responder and non-responder).

**Figure 4 microorganisms-11-01305-f004:**
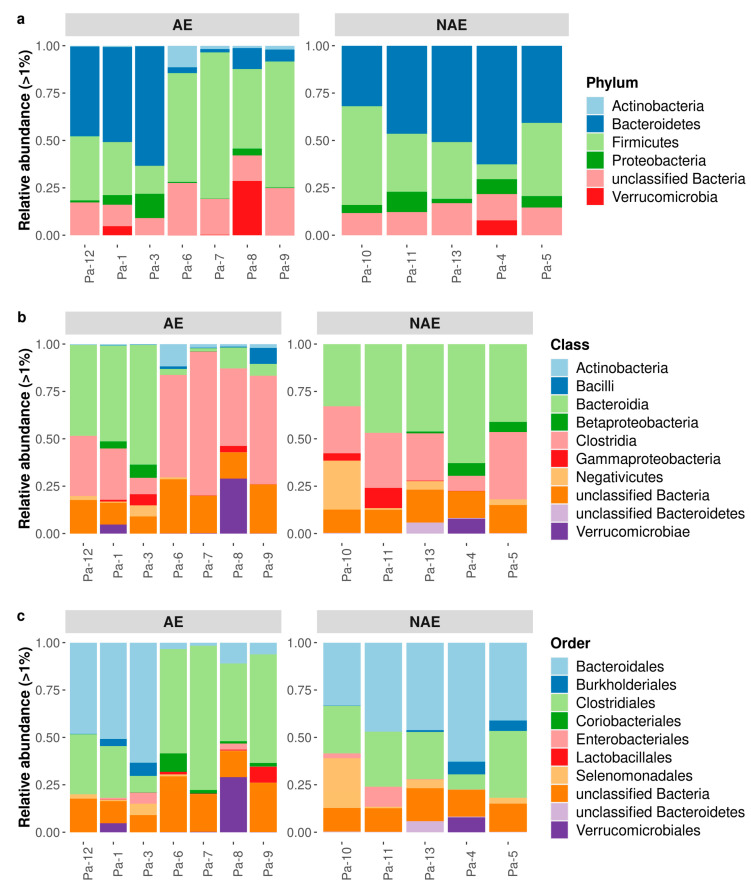
Taxonomic composition and diversity of fecal bacterial communities for each patient who experienced an irAE (AE) or did not (NAE). The most represented phyla (**a**), classes (**b**), and orders (**c**) identified in the study groups are shown with relative abundance. Only taxa whose relative abundance was >1% in at least one group were included.

**Figure 5 microorganisms-11-01305-f005:**
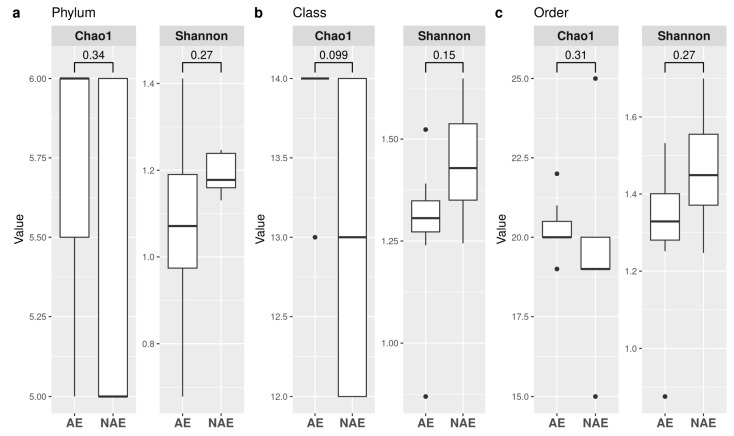
α-diversity according to the experience of an irAE (AE) or not (NAE). Observed richness and Shannon indices are presented at the taxonomic level of phylum (**a**), class (**b**), order (**c**). Significant (*p* < 0.05) comparisons between AE and NAE are indicated in the boxplot.

**Figure 6 microorganisms-11-01305-f006:**
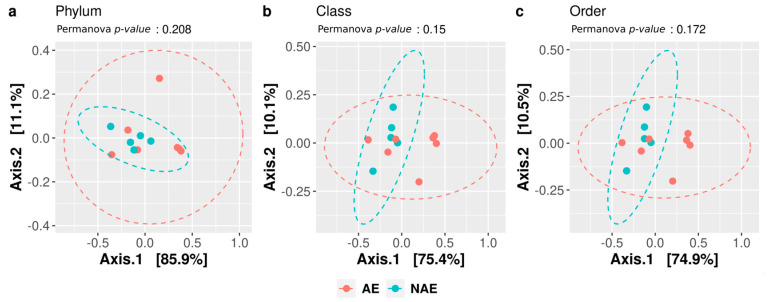
Beta-diversity according to the experience of an irAE (AE) or not (NAE). The microbiota distances were evaluated through the Bray-Curtis dissimilarity matrix at the taxonomic level of phylum (**a**), class (**b**), and order (**c**) and visualized through principal coordinate analysis (PCoA). Each point represents the microbiota composition of one sample.

**Table 1 microorganisms-11-01305-t001:** Patients’ characteristics. B-cell receptor inhibitors used to treat patients are shown in the BCRi. Mutated (mut.) genes, unmutated (unmut.) genes, deletions of chromosomes and the trisomy recorded are listed in genomic features. irAEs are the immune-related adverse events experienced by each patient. Non-responder (NR) and responder (R) patients are shown in the outcome.

ID Patient	Age (YO)	Sex	BCRi	Genomic Features	Treatment Lines	Antibiotic Intake	irAEs	Outcome
01	40–50	M	Ibrutinib	TP53 mut., IGVH unmut.	3°	No	Yes	NR
03	60–70	M	Idelalisib	TP53 mut., IGVH unmut.	2°	Yes	Yes	R
04	60–70	F	Ibrutinib	TP53 mut., IGVH unmut., del(13q), del(17p)	2°	Yes	No	R
05	60–70	F	Ibrutinib	TP53 mut., IGVH unmut.	2°		No	R
06	60–70	M	Idelalisib	TP53 mut., IGVH unmut., del(17p), Trisomy 13	2°	Yes	Yes	R
07	60–70	F	Idelalisib	TP53 mut., IGVH unmut., del(13q)	4°	Yes	Yes	R
08	60–70	F	Idelalisib	TP53 mut., IGVH unmut., del(13q)	2°	Yes	Yes	R
09	60–70	M	Ibrutinib	TP53 mut., IGVH unmut., del(17p)	5°	Yes	Yes	NR
10	60–70	M	Ibrutinib	IGVH unmut., del(13q)	4°	Yes	No	R
11	70–80	F	Ibrutinib	TP53 mut., IGVH unmut., del(17p)	1°	No	No	R
12	70–80	F	Idelalisib	IGVH unmut., del(13q)	2°	No	Yes	R
13	60–70	M	Ibrutinib	IGVH unmut., Trisomy 13	3°	Yes	No	R

**Table 2 microorganisms-11-01305-t002:** Bacterial taxa found only in the AE or NAE group.

	AE Group	NAE Group
Phylum	Actinobacteria	-
Class	Actinobacteria	unclassified Bacteroidetes
	Bacilli	
Order	*Coriobacteriales*	unclassified Bacteroidetes
	*Lactobacillales*	
Family	*Coriobacteriaceae*	*Veillonellaceae*
		*Prevotellaceae*
Species	*Akkermansia muciniphila* ATCC BAA-835	*Bacteroides eggerthii*
	*Blautia luti*	*Veillonella spp* ICM51a
	*Ruminococcus obeum*	*Faecalibacterium prausnitzii*
	*Barnesiella spp* EBA4-14	*Bacteroides ovatus*
		*Butyrivibrio fibrisolvens*
		*Akkermansia muciniphila*
		*Bacteroidaceae bacterium* Smarlab 3301643
		*Parabacteroides distasonis* ATCC 8503

## Data Availability

All the data supporting the findings of this study can be found within the article.

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
