# Peer review of "Gut Microbiota and B Cell Receptor (BCR) Inhibitors for the Treatment of Chronic Lymphocytic Leukemia: Is Biodiversity Correlated with Clinical Response or Immune-Related Adverse Event Occurrence? A Cross-Sectional Study"

_microorganisms, 2023, doi:10.3390/microorganisms11051305_

Round 1

Reviewer 1 Report

line 88 vs 96: line 88 states that all patients received BCRi therapy for at least 12 months, yet line 96 defines NR as patients with therapy discontinuation within 12 months. It is not clear to me how long NR received therapy.

A table with treatment details of all included patients might help.

line 126: this sentence seems to be truncated

line 129: please provide the used RDP database release version

line 130 vs line 133: Please explain the difference between these 2 prevalence cutoff values

line 158: the sentence is truncated

paragraph 3.1: needs some language checking

Figure 4: why does Fig 4b depict single patients? This is not hinted anywhere in the text. Patient 2 is missing? Which of the samples are from NR patients?

Overall:

The manuscript contains a number of truncated sentences and formatting issues as well as misplaced subfigure labels.

While this possibly happened during pdf-conversion (which should be checked by the editors), in its present state it appears that no or incomplete proof reading was performed, which reduces confidence in the presented data.

The demographic data listed in lines 98-101 are no further mentioned. Is there a correlation of GM composition with any of these parameters?

The NR group is too small and only received ibrutinib. It would be interesting to see a comparison of R vs NR only within ibrutinib-treated patients or a comparable group size and treatment regime of R and NR patients.

all figures: need some overhaul. Labels, legend, axis are barely readable, width of the bars provides no information and should be reduced, plotting area too big and empty, should be more compact.

all figures: depicting composition of each single patient (as in fig 4b) would be more informative than mean values

The discussion is well written and draws plausible conclusions. However, it also admits the study's main drawback, which is its small sample size.

While the initial hypothesis (line 259) cannot be confirmed in this study, it is still of value to the CLL research community if the relationship between GM composition and patient characteristics is clearly presented.

Hence, I would like to ask the authors to:

1. if possible add some more NR patient samples to make up for the high inter-patient variability

2. if this is not possible then only compare ibrutinib treated samples

3. show composition plots for each single patient in each figure (like in Fig 4b)

4. add demographic and treatment annotations to the graphs. R pheatmap package is very convenient for this

5. overhaul all figures, increase labels and legends, compact graphs and remove uninformative plotting area

6. check language and formatting throughout the manuscript

Reviewer 2 Report

The paper is well written even if it needs some language improvements for a better understanding.

The authors analyzed the GM biodiversity in CLL patients, interestingly a very large difference in alpha and beta diversity did not emerge but a variability in the distribution of Bacteroidia and order Bacteroidales in the responders groups was noted. Conversely, as regard of Firmicutes and Bacteroidetes a reversal in their relationship in group AE was pointed out

It is important to consider how the GM can have its role in cases of a disease in which the signaling of the BCR and all that follows is altered, in this regard it would be interesting to correlate the IGHV mutational status, looking for some more representative B-CLL cells subpopulations (clinically dominant clone) during the prognosis of the patients, since a correlation between the mutational status and the prognosis of the patients is now well known (i.e. PMID: 32483301): perhaps this is also due to the role of the different antigenic stimulation due to GM components.

However, the study seems valid to me despite the small number of patients - n12.

I recommend a broader language revision and the acceptance in Microorganisms journal. 

Round 2

Reviewer 1 Report

All issues have been addressed and the manuscript was edited accordingly.

I think the data presentation (especially graphics) have significantly improved and the work can be published in its present form.